# Adherent Bacteria and Parasiticidal Secretion Products of Human Cervicovaginal Microbiota-Associated *Lactobacillus gasseri* Confer Non-Identical Cell Protection against *Trichomonas vaginalis*-Induced Cell Detachment

**DOI:** 10.3390/ph15111350

**Published:** 2022-10-31

**Authors:** Bénédicte Pradines, Séverine Domenichini, Vanessa Lievin-Le Moal

**Affiliations:** 1Inserm, UMR-S 996 Inflammation, Microbiome and Immunosurveillance, University Paris-Saclay, 91400 Orsay, France; benedicte.pradines@universite-paris-saclay.fr; 2Unité Mixte de Services, Ingénierie et Plateformes au Service de l’Innovation Thérapeutique (UMS-IPSIT), Microscopy Facility, Université Paris-Saclay, 91400 Orsay, France; severine.domenichini@universite-paris-saclay.fr

**Keywords:** *Trichomonas vaginalis*, motility, adhesion, cytotoxicity, vaginal microbiota, *Lactobacillus gasseri*, released products, parasiticidal activity, HeLa cells

## Abstract

*Trichomonas vaginalis*, a protozoan parasite specific to the human genital tract, is one of the most common sexually transmitted pathogens. Its pathogenicity is strongly associated with its expression of a broad array of proteases triggering cytotoxic effects in host epithelial cells. Vaginal microbiota-associated *Lactobacillus*, including those of *L. gasseri* in particular, can counteract *T. vaginalis* pathogenesis, but the mechanisms involved have yet to be clarified. *T. vaginalis* strain G3 (*Tv* G3) cytotoxicity was assessed by examining cell morphology, cell detachment, and fluorescent labeling of the F-actin cytoskeleton and immunolabeling of vinculin-position focal adhesions (FAs) by confocal laser scanning electron microscopy on confluent cervicovaginal epithelial HeLa cell monolayers. The inhibitory effects of bacterial cells and secreted products of *L. gasseri* ATCC 9857 and KS 120.1 on the *Tv* G3 viability and parasite deleterious effects on HeLa cells were investigated. Pre-adhering *L. gasseri* cells delayed but did not inhibit *Tv* G3-induced cell detachment, F-actin cytoskeleton disorganization and the disappearance of vinculin-positive focal FAs. *L. gasseri* KS 120.1 secretion products had a rapid parasiticide activity by killing time- and concentration-dependent *Tv* G3 parasites after direct contact. By killing *Tv* G3 parasites already associated with the epithelial cells, secretion products have abolished parasite-induced cell detachment. Our findings suggest that vagina microbiota-associated *L. gasseri* creates a physical barrier and exerts pharmacological-type mechanisms to counteract the deleterious cytotoxic effects of *T. vaginalis.*

## 1. Introduction

Trichomoniasis is the most common nonviral sexually transmitted infection worldwide, with about 276 million new cases in women and men each year [1]. In women, trichomoniasis leads to infertility, endometritis, and premature birth and a low birth weight if contracted during pregnancy. Infection in men is usually self-limiting, and often asymptomatic, but may be associated with urethritis, prostatitis, epididymitis, reduced sperm function, and infertility. Trichomoniasis increases the risk of infection with HIV and HSV-2 [2], and has been shown to be associated with an increase in the risk of bacterial vaginosis [3,4] and vulvovaginal candidiasis [5]. Moreover, *T. vaginalis* infection has also been linked to pelvic inflammatory disease and cervical cancer [6]. *T. vaginalis* is susceptible to nitroimidazoles, but treatment failure and relapse are common and related, in some cases, to the emergence of resistant strains [7].

*T. vaginalis* is a member of the Parabasalia group of single-celled eukaryotes within the Excavata clade. There are two cellular forms of infectious *T. vaginalis*. The first is a free-swimming flagellate ellipsoid trophozoite. These trophozoites flatten and spread out over the vaginal epithelium, and are then converted to an ameboid form, maximizing the cell-to-cell contact to ensure efficient, stable establishment of the parasite [8,9]. The genome sequence of *T. vaginalis* has several unique characteristics, including a large size (~160 megabases), with extensive gene duplication and the presence of families of transposable elements [10,11]. The pathogenesis of trichomonal infections remains incompletely understood, but is clearly multifactorial. *T. vaginalis* colonizes the host vaginal epithelium by interacting with human vaginal epithelial cells (VECs), which produce membrane-associated adhesion factors [12,13,14,15,16,17,18,19] and proteases [20,21,22,23,24,25,26]. Deleterious effects on the host are triggered by the production, by *T. vaginalis*, of a highly complex degradome comprising 446 putative peptidase genes encoding enzymes of the five major classes: cysteine, metallo-, serine, threonine, and aspartic proteinases [10]. Mostly through its membrane-associated cysteine proteases, *T. vaginalis* promotes contact-dependent cytolysis [18,24,27], cell monolayer destruction [21,22,26,28], cell detachment [20,25,29], and apoptosis [23,30,31,32] in hVECs and diverse vaginal epithelial cells. The membrane-associated cysteine proteases TvCP39 [26] and TvCP65 [21], and the membrane-associated rhomboid serine protease TvROM1 [25], in particular, have been demonstrated to participate in some of these effects. *T. vaginalis* strains also exert contact-independent deleterious effects on cells, including cytolysis mediated by secreted products [33], cell monolayer destruction mediated by the TvCP2 cysteine protease [34] and the TvGP63 metalloprotease [28], cell detachment mediated by a cell-detachment factor [35] and the metalloprotease TvGP63 [30], and apoptosis induced by the TvCP2 cysteine protease [31] and the TvGP63 metalloprotease [36].

Vaginal eubiosis is associated with a microbiota consisting of community state types I, II III, IV and V, with types I, II, II and V dominated by *L. crispatus*, *L. gasseri*, and *L. jensenii* [37,38]. In trichomoniasis, the cervicovaginal microbiota of the woman is characterized by low levels of health-associated *Lactobacillus* species and a high proportion of *Mycoplasma*, *Parvimonas*, *Sneathia*, and other anaerobes [3,39]. Some vaginal and urinary tract microbiota-associated strains of *L. crispatus* and *L. gasseri* seem to play a key role in frontline defenses against many types of bacterial urogenital infection [40,41]. However, there has been little investigation of the strains of cervicovaginal microbiota-associated *Lactobacillus* capable of antagonizing *T*. *vaginalis* pathogenesis [42,43,44]. In this study, we analyzed the antagonistic effects of strains of *L. gasseri*, one of the major species of the cervicovaginal microbiota of healthy women [37,38], on the deleterious cell cytotoxic effects of *T. vaginalis*. We found that *Lactobacillus* strains combat the colonization of the epithelia by microbial pathogens via an adhesion-dependent mechanism, inhibiting the growth of microbial pathogens through the production of bacteriostatic bacteriocins, and killing microbial pathogens through the secretion of microbicidal metabolites and products [45]. We therefore chose to study two *L. gasseri* strains differing in terms of these activities, as vaginal microbiota-associated test strains. *L. gasseri* strain ATCC 9857 inhibits the colonization of vaginal epithelial cells by the genitourinary tract-associated bacterial pathogens *Prevotella bivia*, *Gardnerella vaginosis* and *Neisseria gonorrhoeae* [46,47,48], and the *T. vaginalis* parasite [43,44], through the action of its cells and a cell-surface aggregation-promoting factor [43,44]. *L. gasseri* strain KS 120.1 secretes products with bactericidal action against *P. bivia* and *G. vaginalis* [46,47]. We demonstrate here that these strains provide different levels of protection against the deleterious cell detachment mediated by the proteases of *T. vaginalis*. The bacterial cells exert an adhesion-dependent inhibition activity that yields partial protection, with total protection achieved by the generation and secretion of strain-specific products with anti-trichomonadal activity.

## 2. Results

### 2.1. Mechanism of Tv G3-Induced Cell Detachment in Human Cervicovaginal Epithelial HeLa Cells

HeLa cells forming a confluent monolayer attached to the glass coverslips displayed time-dependent detachment after infection with *Tv* G3 parasites (Figure 1A). As expected, given that the cell detachment process induced by *T. vaginalis* is known to be mediated by cysteine, serine, and metallo (proteases- [49,50,51], the infection of HeLa cell monolayers with *Tv* G3 parasites in the presence of a combination of cysteine (E64), serine (PMSF), and metallo- (1,10-PT) protease inhibitors completely prevented cell detachment (Figure 1B). We then used DIC microscopy to investigate changes in cell shape and cell-to-cell contacts during the cell-detachment process, over a time course of infection. High-magnification micrographs (Figure 1C) and quantification (Figure 1D) revealed major time-dependent changes in cell morphology, characterized by a shift to a highly rounded shape and the opening of deep rifts at points of cell-to-cell contact. The shape of epithelial cells is defined principally by the actin cytoskeleton, a complex structure consisting of F-actin filaments organized into a stress fiber array by numerous accessory proteins [52]. We therefore visualized the F-actin networks of control unexposed and *Tv* G3-exposed cells, by direct labeling with fluorescein phalloidin. Confocal laser scanning microscopy (CLSM) (Figure 1E) and the determination of F-actin relative fluorescence intensity (RFI)/cell (Figure 1F) over the time course of infection demonstrated a dramatic disappearance of the F-actin-positive straight stress fibers present in control of unexposed cells during the first three hours of *Tv G3* infection. Vinculin is an actin-binding protein present in the focal adhesions (FAs) and involved in cell-substrate interactions; it also operates as a cadherin/β-catenin-binding protein in cell-to-cell contacts [53]. We therefore visualized vinculin by indirect immunolabeling in control unexposed and *Tv* G3-exposed cells. CLSM micrographs demonstrated that there were far fewer vinculin-positive FAs in *Tv* G3-exposed cells than in unexposed cells (Figure 1G). The determination of vinculin RFI/cell (Figure 1H) and counts of vinculin-positive FAs/cell (Figure 1I) over the time course of infection demonstrated that the loss of vinculin-positive FAs began in the sixth hour of infection. Moreover, measurements of the size of the vinculin-positive FAs revealed that all unexposed cells contained large numbers of large vinculin-positive FAs, whereas the *Tv* G3-exposed cells had smaller numbers of vinculin-positive FAs that were also much smaller in size (Figure 1J).

### 2.2. Pre-Adherent L. gasseri Bacterial Cells Delay Tv G3-Induced Cell Detachment

Phunkan et al. [44] demonstrated that the adhesion of *T. vaginalis* F1623, G3 and BR7C2 parasites to hVECs is strongly inhibited if these cells are already colonized by *L. gasseri* ATCC 9857 bacteria. We investigated whether the adhesion of *Tv* G3 trophozoites to confluent HeLa cell monolayers was inhibited by the precolonization of the cells with *L. gasseri* KS 120.1. We used *L. gasseri* ATCC 9857 cells as a positive control. Similar, highly significant decreases in the adhesion of *Tv* G3 trophozoites to cell monolayers were observed for cells precolonized with *L. gasseri* ATCC 9857 and cells precolonized with *L. gasseri* KS 120.1 (79 ± 25% and 81 ± 27% decreases, respectively, Figure 2A). We then investigated the possible consequences of this inhibition for the protection of the cells against *Tv* G3-induced cell detachment. DIC images and quantification (Figure 2B,C, respectively) showed that cell monolayers already colonized with *L. gasseri* ATCC 9857 or KS 120.1 were partially protected, as *Tv* G3-induced cell detachment was not abolished, but occurred 12 h later than in *Tv* G3-exposed cell monolayers not previously colonized with *L. gasseri*.

We then considered the changes to the cell surface, F-actin cytoskeleton, and vinculin-positive FAs over the time course of infection in cell monolayers colonized with *L. gasseri* ATCC 9857 or KS 120.1 before infection with *Tv* G3. DIC micrographs and quantification revealed that cells presenting morphological changes, such as rounding and the opening up of cell-to-cell contacts did not appear until hour 12 of infection in *Lactobacillus*-precolonized *Tv* G3-exposed cells (Figure 3A,B), whereas these changes were already visible during the first three hours of infection in *Tv* G3-exposed cells not previously colonized with *L. gasseri* (Figure 1C,D). Consistently, disappearance of the F-actin cytoskeleton of the cell was not observed until hour 12 of infection in *Lactobacillus*-precolonized *Tv* G3-exposed cells (Figure 3C,D), whereas it was observed during the first three hours of infection in non-pre-colonized *Tv* G3-exposed cells (Figure 1E,F). An analysis of *Lactobacillus*-precolonized *Tv* G3-exposed cells revealed that the decreases in the number and size of vinculin-positive FAs began in hour 18 of infection (Figure 3E–G), whereas these changes were visible as early as hour 6 of infection in non-precolonized *Tv* G3-exposed cells (Figure 1G–I).

### 2.3. Anti-Trichomonadal Activity of Secreted Products of L. gasseri 

Secreted products of *L. gasseri* strain KS 120.1 other than lactic acid have been reported to display bactericidal activity against the vaginosis-associated species *P. bivia* and *G. vaginalis* [46,47]. However, anti-trichomonadal activity has never before been reported for vaginal microbiota-associated *Lactobacillus*. We performed a preliminary experiment to determine whether products secreted by *L. gasseri* ATCC 9857 and KS 120.1 had anti-trichomonadal activity against the *Tv* G3 parasite. In *T. vaginalis,* swimming motility is considered to be a sign of parasite viability [54]. We therefore examined and quantified the swimming motility of untreated and treated axenic *Tv* G3 trophozoites by time-lapse imaging over a time course of direct contact. We found that *L. gasseri* KS 120.1 CFCS containing products secreted by these cells abolished parasite motility in a time- and concentration-dependent manner (Figure 4A,B, respectively). CFCS-induced parasite death was confirmed by assessing trypan blue exclusion after four hours of treatment (not shown). The *L. gasseri* ATCC 9857 CFCS containing secretion products did not inhibit parasite swimming motility (Figure 4A). Parasite swimming motility was not affected by acidity (MRS at pH 4.5) or by the principal metabolite of *Lactobacillus*, protonated lactic acid, at the concentration present in *L. gasseri* KS 120.1 CFCS (MRS pH 4.5 containing 70 mM DL-lactic acid) (Figure 4A). Thus, a product other than protonated lactic acid secreted by *L. gasseri* KS 120.1 exerted strong, rapid anti-trichomonadal effect on coming into direct contact with the parasite.

### 2.4. The Anti-Trichomonadal Activity of L. gasseri Secretion Products Abolishes the Cell-Detachment Effect in Cells Pre-Exposed to Tv G3 Parasites

We then investigated whether the products secreted by *L. gasseri* KS 120.1 that displayed anti-trichomonadal activity in direct contact with *Tv* G3 parasites could eradicate parasites already associated with confluent HeLa cell monolayers. The exposure of cell monolayers to undiluted CFCS for periods of more than six hours resulted in cell damage. We therefore used a 10-fold dilution of CFCS, which retained significant levels of anti-trichomonadal activity (Figure 4B). The number of viable adherent *Tv* G3 parasites decreased significantly when the pre-exposed cell monolayers were treated with a 10-fold dilution of *L. gasseri* KS 120.1 CFCS (82 ± 11% decrease in the number of viable cell-associated parasites relative to pre-exposed untreated cell monolayers) (Figure 5A). Consistent with this lack of anti-trichomonadal activity in direct contact observed with ATCC 9857 CFCS and MRS pH 4.5 + 70 mM DL lactic acid (Figure 4), we found that these treatments did not decrease the number of viable *Tv* G3 parasites associated with the cell monolayers (Figure 5A). We then investigated whether the 10-fold dilution of *L. gasseri* KS 120.1 CFCS protected cells against parasite-induced cell detachment. The time-course data demonstrated that all the cells exposed to *Tv* G3 before treatment with *L. gasseri* KS 120.1 CFCS remained attached to the glass substrate (Figure 5B,C). By contrast, as expected given their lack of anti-trichomonadal activity (Figure 4A), the ATCC 9857 CFCS and MRS pH 4.5 + 70 mM DL lactic acid did not decrease *Tv* G3-induced cell detachment (Figure 5B). Immunolabeling of the F-actin cytoskeleton revealed that the cells exposed to *Tv* G3 before treatment with *L. gasseri* KS 120.1 CFCS retained an almost normal cytoskeleton (Figure 6A,B). An examination of the vinculin-positive FAs of these cells revealed the conservation, throughout the time course, of a large number of large vinculin-positive FAs distributed throughout the cell cytoplasm resembling those present in unexposed untreated control cells (Figure 6C–F).

## 3. Discussion

By analyzing at the changes in the F-actin cytoskeleton and FA levels over time in *Tv* G3-exposed cervicovaginal epithelial HeLa cells, we were able to gain insight into the mechanism by which *T. vaginalis* exerts its cell-detachment effect. Early in exposition, before the occurrence of cell detachment, the exposed cells became more rounded. Lin et al. [23] also reported a rounding of the surface of Z172 human cervical epithelial cancer cells exposed to *T. vaginalis* strain ATCC 30236. Consistent with the action of the actin cytoskeleton in organizing epithelial cell shape through the creation of membrane tension forces [55], we discovered that while the exposed cells underwent a rapid, dramatic disorganization of the F-actin cytoskeleton characterized by the complete disappearance of straight F-actin stress fibers whilst they were rounding. Furthermore, early in infection, cell detachment was preceded by a deepening and widening of cell-to-cell contacts. A similar change in cell-to-cell contacts has already been described in Z172 cells exposed to *T. vaginalis* ATCC 30236 [23]. We first demonstrated that exposed cells presented major changes in the expression of vinculin-positive FAs. This alteration began before the start of cell detachment and increased over the time course of infection until the large FAs were reduced to very small dots. This finding is of mechanistic importance, because FA-associated vinculin is known to establish a link to E-cadherin/α-actinin/β-catenin at the cell-to-cell junctional domain, and to create a link to actin and talin at the FAs for recognition of the basal adhesion receptor integrins, which play a key role in the basal attachment of cells [53]. As shown here, changes to the FAs induce the detachment of diverse epithelial cells [56,57,58]. We think that the time-dependent changes observed at sites of cell-to-cell contact and vinculin-positive FAs are key cellular alterations by which *T. vaginalis* generates its deleterious cell-detachment effect. The changes to FAs may involve the cleavage of FA-associated proteins by *T. vaginalis* serine proteases. Indeed, several bacterial serine proteases, such as the secreted serine protease autotransporter toxin Sat of uropathogenic *Escherichia coli* (*E. coli*) [59] and the serine protease EspC of enteropathogenic *E. coli* [60], have been shown to dissemble the F-actin cytoskeleton and to alter the expression FA-associated proteins, thereby promoting cell detachment. Nevertheless, a role for *T. vaginalis* cysteine proteases cannot be excluded, because the cysteine protease calpain has already been demonstrated to cleave vinculin [61], decreasing the association of vinculin with FAs [62].

Urogenital microbiota-associated *Lactobacillus* strains are essential for the maintenance of homeostasis in the vagina and urinary tract [40,41]. *T. vaginalis* and vaginosis-associated bacterial pathogens have deleterious effects on vaginal microbiota-associated *Lactobacillus,* causing a loss of these beneficial bacteria [3,4,39,63]. Our results provide additional knowledge about the protective role of vaginal microbiota-associated *Lactobacillus* against the deleterious effects of *T. vaginalis* on the vaginal mucosa. Prior incubation of human amnion-derived WISH cells with *L. fermentum* has been demonstrated to inhibit *T. vaginalis* adhesion [42]. Moreover, Phukan et al. [43,44] reported that *L. gasseri* ATCC 9857 bacterial cells with a high adhesion capacity can inhibit the association of the *Tv* G3, B7RC2 and F1623 strains with human vaginal ectocervical CRL-2614 cells. Here, we confirm the inhibitory effect of *L. gasseri* ATCC 9857 bacterial cells on the association of *Tv* G3 with cells and demonstrate that *L. gasseri* KS 120.1 bacterial cells have a similar inhibitory effect. We also report that the cell-detachment effect of *Tv* G3 was delayed rather than abolished in HeLa cell monolayers colonized with *L. gasseri* ATCC 9857 or KS 120.1 bacterial cells before infection with the parasite. A similar time shift in cytotoxic effect has been observed in HeLa cell monolayers precolonized with vaginal microbiota-associated *L. gasseri* ATCC 9857 or KS 120.1 cells, or *L. crispatus* CTV-05 cells, before treatment with the secreted vaginolysin of *G. vaginalis* and the Sat toxin of UPEC [47]. We think that this retarded effect may reflect the inability of adherent *Lactobacillus* cells to form a sufficiently dense bacterial biofilm to create an impermeable physical barrier in our experimental conditions. Moreover, the adherent *Lactobacillus* cells are probably unable to block the diffusion of secreted *T. vaginalis* proteases known to be involved in the parasite cell-detachment effect [28,30,33,34,35].

We found that *L. gasseri* KS 120.1 CFCS containing secreted products had a rapid, strong pharmacological-type anti-trichomonadal activity against axenic *Tv* G3 trophozoites. Moreover, we observed that the killing of parasites already present in HeLa cells resulted in the complete protection of cervicovaginal cells against *Tv* G3-induced cell detachment. A similar protective effect against *T. vaginalis* adhesion-dependent cytotoxicity has been observed in hVECs and HeLa cells exposed to parasites treated with metronidazole before cell infection [27,64]. The *L. gasseri* KS 120.1 secreted products involved remain to be identified. *Lactobacillus* strains are known to act by producing strain-specific bacteriocins, and non-strain-specific metabolites, including protonated lactic acid and other heat stable, pH-sensitive non-lactic acid antimicrobial compounds, and phenyllactic acid, a heat-stable, pH-sensitive organic acid with strong bactericidal activity [65,66]. In the absence of cultured bacteria, MRS (pH 4.5) alone and MRS containing DL-lactic acid at pH 4.5 have no anti-trichomonadal effect against *Tv* G3, indicating that acidity and protonated lactic acid are responsible for the anti-trichomonadal effects of *L. gasseri* KS 120.1. Additional studies are required to determine whether the bactericidal and anti-trichomonadal effects of the secreted products of *L. gasseri* strain KS 120.1 are mediated by the same or different molecules.

Probiotics have been defined by the FAO/WHO Panel of Experts as “Live microorganisms that, when administered in adequate amounts, confer a health benefit on the host” [67,68]. As a means of combatting vaginosis-associated bacterial pathogens, several vaginal *Lactobacillus* strains with the required probiotic profile [69] could constitute the basis of a valid alternative to antibiotics for the prevention and treatment of urogenital tract infections [70]. The clinical use of probiotic vaginal *Lactobacillus* strains to treat trichomoniasis has never been reported. Only one clinical report to date has mentioned that metronidazole treatment can be improved by the addition of a *Lactobacillus casei rhamnosus* strain [71]. In this context, *L. gasseri* ATCC 9857 and KS 120.1 are promising probiotic strains for replenishing *Lactobacillus* levels in patients with *T. vaginalis*-induced microbiota dysbiosis because they are highly adhesive [43,44,46,47,48]. Postbiotics have been defined as “preparations of inanimate microorganisms and/or their components that confer a health benefit on the host” [72]. An additional interest of the *L. gasseri* strain KS 120.1 is its potential for use as a postbiotic, given that it generates secreted products with rapid, strong, pharmacological-like anti-trichomonadal activity, protecting ectovaginal cells against the deleterious cytotoxic effect of *T. vaginalis*. Products secreted by *L. gasseri* KS 120.1 also act as pharmacological-like bactericides against *P. bivia* and *G. vaginalis* [46,47] and have a strong protective effect on HeLa cells pre-infected with an UPEC producing the Sat toxin, which triggers cell detachment [47]. These secreted products may have therapeutic effects when delivered directly to the vagina by appropriate vehicles, such as vaginal tablets, capsules, or suppositories [73].

## 4. Materials and Methods

### 4.1. Reagents and Antibodies

The E64 cysteine protease inhibitor, serine protease phenylmethylsulfonyl fluoride (PMSF) inhibitor, and metalloprotease 1,10-phenanthroline (1,10-PT) inhibitor were purchased from Sigma-Aldrich (Saint-Quentin-Fallavier, 38070, France). Fluorescein phalloidin was obtained from Molecular Probes (Thermo Fisher Scientific, 91140 Les Ulis, France). Mouse anti-vinculin antibody (clone hVIN-1) was obtained from Sigma-Aldrich. Appropriate secondary antibodies were purchased from Jackson Immunoresearch Laboratories, Inc. (West Grove, PA, USA) and Molecular Probes Inc. (Thermo Fisher Scientific, Invitrogen Life Technologies, San Diego, CA, USA). 

### 4.2. T. vaginalis Strain G3

We used the metronidazole-sensitive *T. vaginalis* reference strain G3 (ATCC PRA-98™; Taxonomy ID: 412133) (*Tv* G3), the genome of which has already been sequenced [10]. This strain was stored in liquid nitrogen supplemented with 6% dimethyl sulfoxide (DMSO). The strain was grown axenically at 35 °C in Diamond’s trypticase-yeast extract-maltose (TYM) medium [74] supplemented with 10% horse serum, with subculturing every two days. For assays, trophozoites were harvested in mid-exponential growth phase. The number of parasites per milliliter was determined under the microscope, with a hemocytometer (Kova^®^ Glasstic^®^ Slide 10, Hycor Biomedical, Garden Grove, CA, USA).

### 4.3. Lactobacillus Strains

*L. gasseri* strain ATCC 9857 was obtained from the American Type Culture Collection (Manassas, VA, USA). The human vaginal microbiota-associated *L. gasseri* strain KS 120.1 was kindly provided by ProbioSwiss (Zurich, Switzerland). All *Lactobacillus* strains were grown in De Man, Rogosa and Sharpe (MRS) broth (Gibco, Thermo Fisher Scientific, 67403 Illkirch, France) for 18 h at 37 °C, under an atmosphere containing 5% CO_2_-95% air. Isolated *Lactobacillus* cells and cell-free culture supernatants (CFCSs) were obtained by centrifuging the *Lactobacillus* cultures (18 h, 10^9^/CFU/mL) at 10,000× *g*, for 30 min at 4 °C. The separated bacterial cells were washed three times with sterile MRS and resuspended in fresh MRS. CFCSs were passed through a sterile 0.22 mm-pore Millex GS filter unit (Millipore, Molsheim, France), and a colony count assay was performed to check that there were no bacterial cells in the CFCSs.

### 4.4. Determination of Anti-Trichomonadal Activity

Time-dependent direct anti-trichomonadal effect was assessed by determining the swimming motility of axenic *Tv* G3 trophozoites. We incubated 500 µL of trophozoite-containing TYM medium (final concentration: 10^8^ parasites/mL) at 37 °C in the presence or absence of 500 µL CFCS for the times indicated. At the end of the incubation period, samples of the incubation medium were collected and examined by differential interference contrast (DIC) microscopy, for the visualization and quantification of motile trophozoites [75]. About 50 µL of incubation medium was pressed between coverslips to form a layer about 50 µm thick, and this pressed mixture was immediately examined by phase-contrast microscopy with an inverted AxioObserver Z1_Colibri microscope (Zeiss, Germany) equipped with an AxioCam MRm charge-coupled-device (CCD) camera (6.45-µm-pixel size) and an Achroplan 10×/0.25 NO dry objective lens at 23 °C (temperature maintained by air conditioning). The results are expressed in arbitrary units (AU): 2.0 AU for 100% motile trophozoites, 1.5 AU for ~75% motile trophozoites, 1.0 AU for ~50% motile trophozoites, 0.5 AU for ~25% motile trophozoites, 0 AU for 100% non-motile trophozoites. Parasite viability was also estimated by trypan blue exclusion, with a hemocytometer. Each assay was conducted at least three times, in duplicate.

### 4.5. Culture of Human Cervicovaginal Epithelial HeLa Cells

Cells were grown to subconfluence in RPMI 1640 with L-glutamine (Thermo Fisher Scientific, Invitrogene Life Technologies) supplemented with 10% heat-inactivated fetal calf serum (FCS; Boehringer Ingelheim, 75644, Paris, France) at 37 °C, under an atmosphere containing 5% CO_2_ [47]. Cell monolayers were used for post-confluence (7 days in culture) infection assays. Before processing, the cells were washed twice with phosphate-buffered saline (PBS).

### 4.6. Cytoadhesion Assay

Confluent HeLa cell monolayers were washed twice with sterile phosphate-buffered saline (PBS) and exposed to *Tv G3* trophozoites (10^7^ parasites/mL) in RPMI 1640 at 37 °C for 30 min under an atmosphere containing 5% CO_2_-95% air. We assessed the ability of *Lactobacillus* cells to inhibit *T. vaginalis* cytoadhesion, by incubating the cell monolayers for 3 h at 37 °C with *Lactobacillus* cells (5 × 10^7^ CFU/mL bacteria) and then washing three times with sterile PBS to remove non-adhering bacteria [47]. The cells were then incubated with *T. vaginalis* trophozoites (10^7^ parasites/mL) for 30 min at 37 °C. The plates were washed five times with sterile PBS to remove non-adhering parasites and the cell monolayers were treated with trypsin (0.25%) at 37 °C to obtain counts of viable adherent parasites. The number of viable parasites was determined in aliquots of the trypsin-treated samples, by obtaining counts of parasites displaying trypan blue exclusion with a hemocytometer under a microscope. Each assay was performed in duplicate, with at least three successive cell passages.

### 4.7. Cell-Detachment Assay

We monitored the inhibition of *T. vaginalis*-induced cell detachment in conditions of protection by lactobacilli, by colonizing confluent HeLa cell monolayers with *L. gasseri* bacterial cells (5 × 10^7^ CFU/mL) for 3 h at 37 °C under an atmosphere containing 5% CO_2_-95% air and then washing them three times with sterile PBS to remove the non-adherent bacteria [47]. The cells were exposed to trophozoites (10^7^ parasites/mL) at 37 °C for the times indicated, and the plates were then washed five times with sterile PBS to remove non-adherent parasites. For assessments of the inhibition of *T. vaginalis* cytotoxicity by protease inhibitors, the cell monolayers were exposed to trophozoites (10^7^ parasites/mL) for 12 h in the continuous presence or absence of a combination of E-64 cysteine protease inhibitor (100 µM) [31], PMSF serine protease inhibitor [76] and 1,10-PT metalloprotease inhibitor [36].

For assessment of the inhibition of *T. vaginalis* cytotoxicity by *L. gasseri* CFCS, the confluent HeLa cell monolayers were exposed to *Tv* G3 trophozoites (10^7^ parasites/mL) for 30 min at 37 °C under an atmosphere containing 5% CO_2_-95% air and then washed five times with sterile PBS to remove non-adherent parasites. They were then either left untreated or were treated with CFCS or MRS pH 4.5 or MRS pH 4.5 + 70 mM lactic acid for the times indicated. For experiments with *Lactobacillus* CFCS, the CFCS was used at a 10-fold dilution, because cell exposure to undiluted CFCS for more than six hours resulted in cell damage.

Cell detachment was quantified by observation of the cells by DIC microscopy under an Aristoplan phase-contrast microscope (Plan Aprochromat 100×/1.32-0.6 oil objective) (Leitz, Wetzlar, Germany) [47,59]. Possible bias was eliminated by performing image analysis and quantification in a blind manner. At least 10 random microscopic fields per sample were examined for the counting of adherent cells. Quantitative values were exported to Excel for further analysis and graphical representation. Photographic images were resized, organized, and labeled with Adobe Photoshop CS6 software (San Jose, CA, USA).

### 4.8. Fluorescence Labeling

Confluent HeLa cell monolayers were cultured on glass coverslips, which were placed in 24-well TPP tissue culture plates (Dutscher, 9213 Issy-les-Moulineaux, France). For microcopy examinations, specimens were fixed by incubation with 3% paraformaldehyde in PBS for 5 min at room temperature, washed three times with PBS, treated with 50 mM NH_4_Cl in PBS for 10 min to neutralize aldehyde groups and blocked by adding 0.2% gelatin in PBS. Cells were permeabilized by incubation with 0.2% Triton X-100 in PBS for 4 min at room temperature and washed three times with PBS. The F-actin cytoskeleton was labeled directly with fluorescein phalloidin (1/200) [59]. Vinculin was immunolabeled with mouse anti-vinculin antibody (1/100) and an appropriate rhodamine-conjugated secondary antibody [59]. The cells were mounted in Dako fluorescent mounting medium (DAKO) and covered with a coverslip.

### 4.9. Confocal Laser Scanning Microscopy (CLMS) and Imaging Analysis

Samples were imaged with a Leica g-STED TCS SP8 inverted confocal microscope (Leica, Germany) equipped with an HC PL APO CS2 63 ×/1.40 oil immersion objective. The instrument was equipped with a WLL laser (excitation wavelengths of 495 nm for green fluorescent protein and 552 nm for rhodamine). The pinhole was set at 1.0 Airy unit, giving an optical slice thickness of 0.89 µm. Twelve-bit digital images were processed with Leica SP8 LAS X software (Version 2.0.1; Leica, Germany). Stacks of confocal laser scanning microscopy (CLSM) images were collected at 0.30 μm intervals along the *z* axis. Relative fluorescence intensity (RFI) was determined for each cell from the analysis of confocal *z*-stack images. The segmented line tool of ImageJ software (version 1.42, NIH, USA) was used to delineate the area of the cell analyzed, and the plot profile tool was used to obtain RFI values in arbitrary units [59]. The number of vinculin-positive FAs/cell was determined by analyzing at least 15 cells per sample with Image J software [59]. The length of the vinculin-positive FAs per cell was determined by analyzing at least 10 cells per sample with Imaris software (version 6.21) (Bitplane AG, Zurich, Switzerland) [59]. Bias was prevented by performing all analyses of images and quantifications in a blind manner. Quantitative values were exported to Excel for further analysis and graphical representation. Photographic images were resized, organized, and labeled with Adobe Photoshop CS6 software (San Jose, CA, USA).

### 4.10. Analysis

The results are expressed as means ± the standard error of the mean (SD). Student’s *t* tests were performed for statistical comparisons.

## Figures and Tables

**Figure 1 pharmaceuticals-15-01350-f001:**
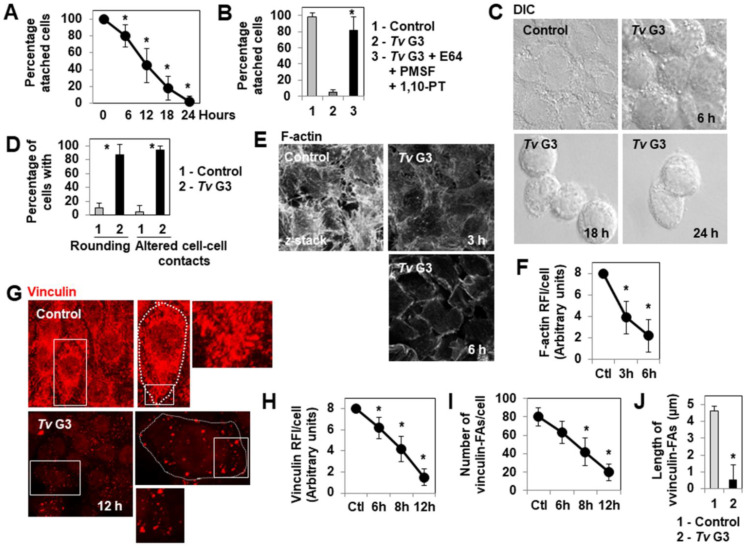
The F-actin network, cell-to-cell contacts, and focal adhesion contacts are altered in a time-dependent manner by *Tv* G3, leading to the detachment of cervicovaginal HeLa epithelial cells. In (**A**), progression of *Tv* G3-induced cell detachment over a time course. In (**B**), *Tv* G3-induced cell detachment with and without a combination of protease inhibitors (the E64 cysteine protease inhibitor at 100 µM, the PMSF serine protease inhibitor at 2 mM, and the 1,10-PT metalloprotease inhibitor at 5 mM) (24 h post-infection (pi)). In (**C**), representative differential interference contrast (DIC) micrographs of cells showing a rounding of the cell surface and alterations to cell-to-cell contacts. In (**D**), percentages of cells displaying rounding and alterations to cell-to-cell contacts (6 h pi). In (**E**), representative confocal laser scanning microscopy (CLSM) images of the F-actin network. In (**F**), decrease in F-actin relative fluorescence intensity (RFI)/cell over time. In (**G**), representative CLSM images of vinculin-positive focal adhesions (FAs). In (**H**), decrease in vinculin RFI/cell over a time course. In (**I**), decrease in the number of vinculin-positive FAs/cell over time. In (**J**), length of vinculin-positive FAs (6 hpi). The micrographs shown are representative of two independent experiments. In (**G**), the white box indicates the cell viewed at high magnification in the adjacent image. The dashed white line delineates the area occupied by the cell cytoplasm. Ctl: Control. Each value shown is the mean ± SD of at least two independent experiments. In (**A**), * *p* < 0.01 for comparisons with T_0_. In (**B**), * *p* < 0.01 for comparisons with *Tv* G3. In (**D**,**F**,**H**–**J**), * *p* < 0.01 in comparisons with Control.

**Figure 2 pharmaceuticals-15-01350-f002:**
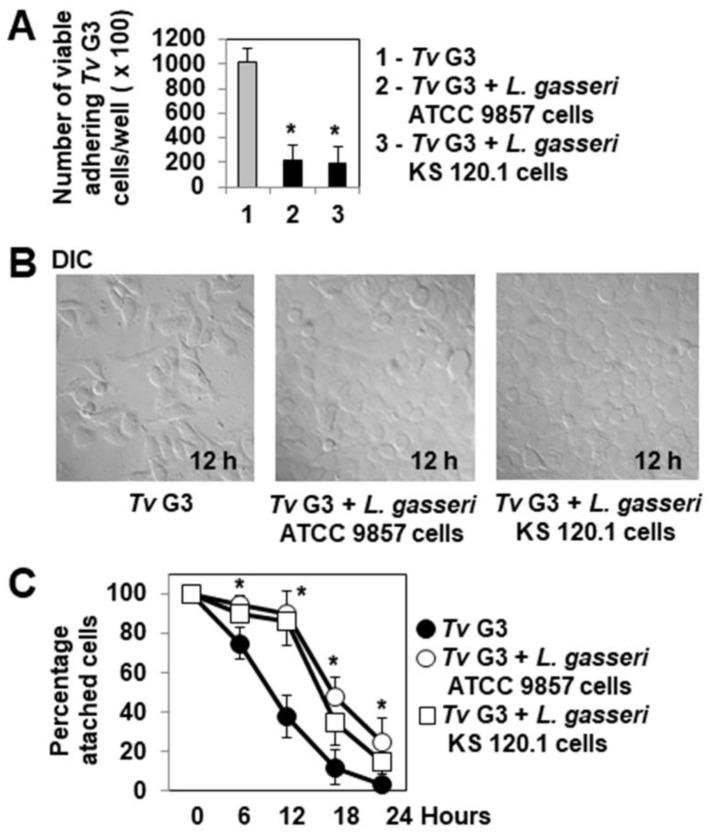
The prior adhesion of *L. gasseri* to cells delays *Tv* G3-induced cell detachment. (**A**) Levels of parasites adhering to HeLa cells with and without prior colonization with *L. gasseri* ATCC 9857 or KS 120.1 cells. In (**B**), representative differential interference contrast micrographs of cells. In (**C**), progression of cell detachment over time for cells with and without prior colonization with *L. gasseri* ATCC 9857 or KS 120.1 cells. The micrographs shown are representative of two independent experiments. Each value shown is the mean ± SD of three separate experiments. In (**A**,**C**), * *p* < 0.01 for comparisons with *Tv* G3.

**Figure 3 pharmaceuticals-15-01350-f003:**
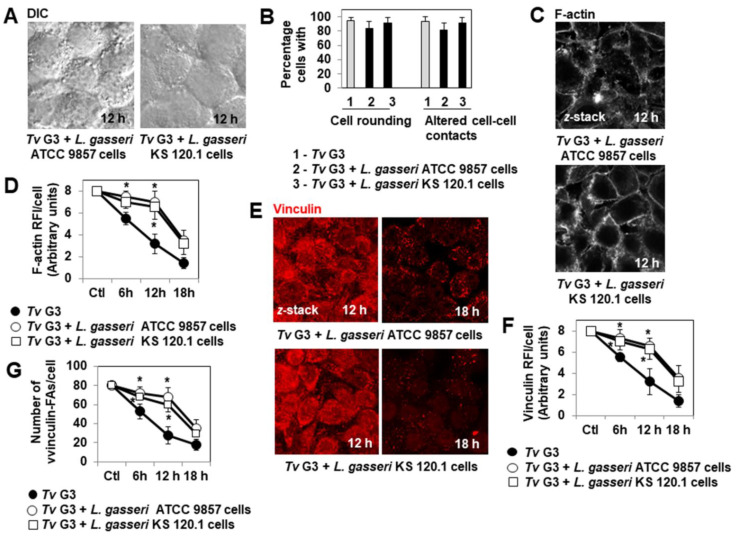
The prior adhesion of *L. gasseri* bacterial cells delays *Tv* G3-induced alterations to the F-actin network, cell-to-cell contacts, and focal adhesion contacts. In (**A**), representative differential interference contrast micrographs of *T. vaginalis*-exposed cells precolonized with *L. gasseri* ATCC 9857 or *L. gasseri* KS 120.1 cells. In (**B**), percentages of cells displaying rounding and alteration to cell-to-cell contacts (12 h post-infection). In (**C**), confocal laser scanning microscopy (CLSM) images of the F-actin network. In (**D**), decrease in F-actin relative immunofluorescence intensity (RFI) over time. In (**E**), representative CLSM micrographs of vinculin-positive FAs. In (**F**), decrease in vinculin RFI/cell over time. In (**G**), decrease in the number of vinculin-positive FAs/cell over time. The micrographs shown are representative of three independent experiments. Ctl: Control. Each value shown is the mean ± SD of three independent experiments. In (**D**,**F**,**G**), * *p* < 0.01 for comparisons with *Tv* G3.

**Figure 4 pharmaceuticals-15-01350-f004:**
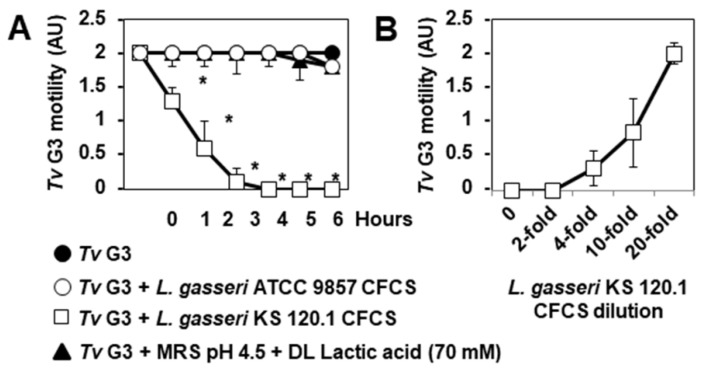
Strain-specific anti-trichomonadal activity of secreted *L. gasseri* KS 120.1 product(s). In (**A**), changes in the swimming motility of untreated and treated axenic trophozoites over the period of direct contact. In (**B**), concentration-dependent inhibition of parasite swimming motility. CFCS: cell-free culture supernatant. Each value shown is the mean ± SD of two experiments. In (**A**), * *p* < 0.01 for comparisons with *Tv* G3.

**Figure 5 pharmaceuticals-15-01350-f005:**
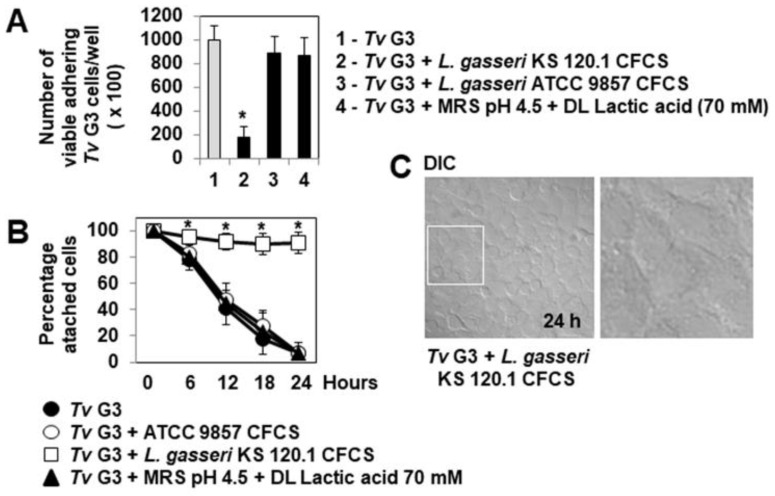
The strain-specific anti-trichomonadal activity of secreted *L. gasseri* KS 120.1 products abolishes *Tv* G3-induced cell detachment. In (**A**), quantification of viable parasites adhering to HeLa cells. In (**B**), progression of cell detachment over the course of infection. In (**C**), representative differential interference contrast micrographs of cells. The micrographs shown are representative of three independent experiments. CFCS: cell-free culture supernatant. Each value shown is the mean ± SD of two experiments. In (**A**,**B**), * *p* < 0.01 for comparisons with *Tv* G3.

**Figure 6 pharmaceuticals-15-01350-f006:**
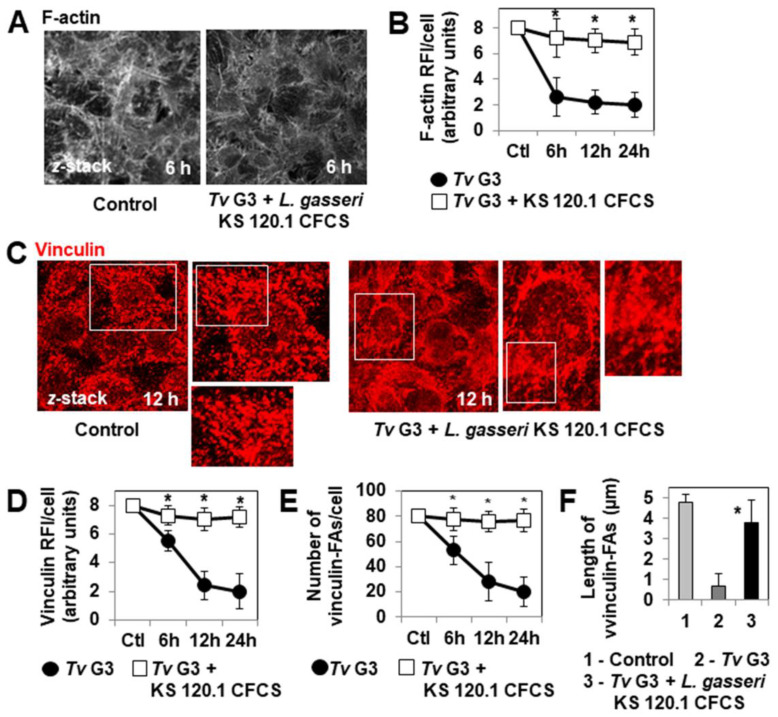
The strain-specific anti-trichomonadal activity of secreted *L. gasseri* KS 102.1 products abolishes *Tv* G3-induced alterations to the F-actin network and vinculin-positive focal adhesions. In (**A**), confocal laser scanning microscopy (CLSM) images of the F-actin network. In (**B**), change in F-actin relative fluorescence intensity (RFI) during the course of infection. In (**C**), representative CLSM images of vinculin-positive FAs. In (**D**), change in vinculin RFI/cell over the course of infection. In (**E**), change in the number of vinculin-positive FAs/cell over the course of infection. In (**F**), lengths of vinculin-positive FAs at 12 h post-infection. The micrographs shown are representative of three independent experiments. CFCS: cell-free culture supernatant. Ctl: Control. Each value shown is the mean ± SD of three independent experiments. In (**B**,**D**–**F**), * *p* < 0.01 in comparisons with *Tv* G3.

## Data Availability

Data reported in this study are contained within the article. The underlying raw data are available on request from the corresponding author.

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
