# Peer review of "Adherent Bacteria and Parasiticidal Secretion Products of Human Cervicovaginal Microbiota-Associated Lactobacillus gasseri Confer Non-Identical Cell Protection against Trichomonas vaginalis-Induced Cell Detachment"

_pharmaceuticals, 2022, doi:10.3390/ph15111350_

Round 1

Reviewer 1 Report

Thank you for the opportunity to review your paper. I thought it was well written and very interesting and has relevant clinical application.

Author Response

We greatly appreciated the constructive comments received. The revised manuscript has been modified and update.

Reviewer 2 Report

This study by Pradines et al. constitutes an important discovery and a potential remedy in the treatment of metronidazole-refractory trichomoniasis cases. The authors show that Lactobacillus gasseri strain KS 120.1 secretes compounds that efficiently kill Trichomonas vaginalis trophozoites. Since L. gasseri is a normal member of the vaginal flora it could be used as a treatment against T. vaginalis infections. The technical quality of the experimental work is also high. I have no relevant queries with the exception of minor corrections regarding style/definition:

Abstract: it should read "Pre-adhering L. gasseri cells delayed but did not inhibit Tv G3-induced....."

Replace "trichomonacide" with "anti-trichomonadal" throughout the manuscript!

Discussion, page 10: the epithelial cells were not "infected" by T. vaginalis because T. vaginalis is a strictly extracellular parasite. Rather write "while being exposed to T. vaginalis.."

Author Response

Please find enclosed the revised manuscript pharmaceuticals-1971852 entitled: "
Cervicovaginal Microbiota-Associated Lactobacillus gasseri Confer Non-Identical Cell
Protection Against Trichomonas vaginalis-Induced Cell Detachment". We greatly appreciated
the constructive comments received. The revised manuscript has been modified in line with
all the suggestions by Reviewer 2.
- In all the text, the terms “trichomonacide” and “trichomonacidal” have been replaced
by “anti-trichomonadal”
- The sentence Lines 31-32: has been modified to read: “Pre-adhering L. gasseri cells
delayed but did not inhibited Tv G3-induced cell detachment, F-actin cytoskeleton
disorganization and the disappearance of vinculin-positive focal FAs.”
- -In all the text, “infected” has been replaced by “exposed”.
Thank you for your cooperation. We hope that you will find that with the changes requested;
the revised manuscript is now suitable for publication in Pharmaceuticals. We look forward to
hearing the outcome of this submission at your earliest convenience.
Yours sincerely,